

# Muography as a new tool to study the historic earthquakes recorded in ancient burial mounds

Hiroyuki K.M. Tanaka[1,2], Kenji Sumiya[3], László Oláh[1,2]

[1]Earthquake Research Institute, The University of Tokyo, 1-1-1 Yayoi, Bunkyo, Tokyo 113-0032, Japan
[2]International Muography Research Organization (MUOGRAPHIX), The University of Tokyo, 1-1-1 Yayoi, Bunkyo, Tokyo 113-0032, Japan
[3]Graduate School of Informatics, Kansai University, 2-1-1 Ryozenji-cho, Takatsuki-shi, Osaka 569-1095, Japan

**Abstract**

Bidirectional muographic measurements were conducted at the Imashirozuka burial mound, Japan. The mound was built in the beginning of the 6th century as a megalithic tomb and was later collapsed after a landslide caused by the 1596 Fushimi Earthquake, one of the largest earthquakes that have occurred in Japan over last few centuries. The measurements were conducted in order to find evidence of this past disaster recorded in this historical heritage sites. As a result, the vertical low-density regions were found at the top of the mound. These regions were interpreted as large-scale vertical cracks that caused the translational collapse process behind the rotational landslide that was already found in the prior trench-survey-based works. These results indicate that there was an intrinsic problem with the stability of the basic foundation of the Imashirozuka mound before the 1596 Fushimi Earthquake.

## 1. Introduction

By expanding our understanding of past large-scale natural disasters, such as tsunami, earthquakes and volcanic eruptions, future hazards can be extrapolated and estimated. However, modern scientific records of these natural disasters only, for the most part, cover events from the last couple of centuries, which have been recorded by scientific instruments only in limited regions throughout the world. However, geographical or



topographical modifications are often physically recorded in the land surface as a result
of such large-scale natural disasters, and correct methodologies can be deciphered to
infer unknown details about these events. For example, a large-scale volcanic eruption
usually creates a large volume pyroclastic flow, which later remains in the geological
stratum as a sedimentation of volcanic products. By applying a geological dating
technique to these past remnants of the eruptions, we can infer the timing and the
magnitude of the past disasters. However, the geological timescale is largely different
from that of human history, and the dating precision by these geochronological
techniques is limited to an order of 100 years. On the other hand, historical studies often
provide records that can be verified with yearly or sometimes, daily precision,
depending on how far back the disaster occurred. Historical information is more
straightforward regarding affected sites and the year or date of the disaster. For example,
this information can come from literature, which describes destruction by earthquakes
or repairs after earthquakes, providing valuable evidence for the location of earthquakes
and the effects brought by these earthquakes. Therefore, if we can combine the
historian's knowledge with the analysis results of these past disaster remnants, historical
records become valuable information which can help to improve the accuracy of these
geological dating techniques by developing into an iteration process. The derivation by
scientists and engineers have been utilized as evidence of earthquakes and which are
later incorporated by historian to evaluate the dates of the events, and vise versa.
Thus far, a combination of geological techniques and historical data have been applied
to historically well-studied objects to fill the gaps in our understanding of the historical
natural disaster record including tsunami (Daly et al., 2019; Dey et al. 2014),
earthquakes (Korjenkov and Mazor, 2003; Guidoboni et al., 1994; Ambraseys et al.
1983) and volcanic eruptions (Elson and Ort, 2018). The data are exploited mostly by
direct excavation of the historic site, and such anatomical techniques (similar in
principle to dissecting bodies to directly view organs within human bodies) allow us to
exploit regional, direct and detailed information; however, not all historical heritage
sites can be accessible and modified in this way. For example, due to the cultural
restriction, it is not always possible to conduct a trench survey to excavate the extant
historical structures such as the ancient monuments or public buildings to obtain the
geological knowledge about the past disaster remnants.   Even when such a style of
investigation is approved, the exploitable information is usually localized. Thus, there is
a need for a non-invasive technique such as surface wave exploration, which would be
conducted to provide a more overall picture of targeted structures to increase the



possibilities of finding more physical evidence of past disasters.

Muography is a technique enabling us to "x-ray" gigantic (hectometric to kilometric)
objects.    The surface of the Earth is constantly bombarded with muons, particles that
have decayed from cosmic rays arriving at the atmosphere from outside our solar
system, and these particles can be utilized as probes for muography.    After traversing
targeted object, remnant muons are tracked with a particle detector located at lower
elevations than the region of interest inside the target. The result is a pattern of the
contrast in the density distribution inside the objects, which is projected on a
2-dimensional plane. Muography has been applied to image the internal structure of
volcanoes (Tanaka et al. 2007; Tanaka et al., 2009; Lesparre et al., 2012; Tanaka et al.,
2014; Olah et al., 2019), cultural heritages including Giza pyramids (Cheops and
Chephren), Egypt, Prambanan temples, Indonesia, Mt Echia, Italy and Santa Maria del
Fiore, Italy (Alvarez et al., 1970; Hanazato and Tanaka, 2016; Tanaka and Ohshiro,
2016; Morishima et al., 2017; Guardincerri et al. 2018; Cimmino et al. 2019), industrial
plants (Tanaka, 2013), and other natural (Tanaka et al., 2011; Olah et al. 2012; Schouten,
2018) and man-made structures (Mohon et al. 2018). Prior works have focused on
searching undiscovered chambers or the total weight of the heritage. Instead, in this
work, we applied muography to study ancient earthquakes for the first time. We focused
on the 1596 Fushimi Earthquake, one of the largest earthquakes that have occurred in
Japan over last few centuries and examine whether the technique of muography can
increase the possibilities of finding more physical evidence of past disasters recorded in
historical heritage sites.

**2. Observation**
Imashirozuka, an imperial burial mound in Japan was chosen as a target of the current
study. In Japan, imperial burial mounds have been well studied and a lot of knowledge
has accumulated. For the current study, this type of the burial mound has the following
advantages to study past earthquakes (Kamai et al., 2008). (A) The construction method
of the imperial mound is well studied by historians and thus, even if the mound has been
damaged by the past earthquakes, the original structure of the mound can be estimated.
(B) The imperial mound was built as a stable object, and thus collapsed areas inside the
mound would be likely to be records of past major earthquakes. (C) The imperial
mounds are in general situated in the urban area. Therefore, the collapsed mounds can
be used as an index to measure the past seismic disasters in urban areas long ago. (D) In
the recent human's history, various kinds of embankments have been built, but its


stability is discussed within the time scale of decades. The collapsed mounds offer us a
unique opportunity of geotechnical discussions within a time scale of centuries. (E) The
construction method of the mound was already well established when they were built.
The mounds built in the same era used the same construction method and thus it is
expected that the mechanical strength is the same. Therefore, the different collapsing
conditions among different mounds located near each other could infer different ground
conditions or different underwater conditions.

Imashirozuka is a keyhole-shaped imperial burial mound that was built in the beginning
of the 6th century in Japan. This burial mound is situated on one of the most active
faults in Japan, which is part of the Rokkou active fault system. This fault system
caused the Great Hanshin Earthquake in 1995.   In 1596, it is thought that this Rokkou
active fault system and the next neighbor fault system called the Arima-Takatsuki
tectonic line were activated at the same time, and one of the largest earthquakes in the
last few centuries, Fushimi Earthquake, (Magnitude 7.25-7.75) occurred (Kamai et al.,
2008). The total length of the Imashirozuka mound is 190 m and the height is 11-12 m.
Although this burial mound was originally built in the triple-layered structure, the top
layer collapsed after a landslide. The collapse occurred more extensively in the northern
part of the mound. The level of damage depends, in general, on the ground motion
during an earthquake, which itself depends on its magnitude and distance from the site.
This extensive collapse is probably due to the existence of the Ai fault line, a part of the
Rokkou active fault system, is located closer to the northern part of the mound.
Currently, Imashirozuka mound consists of a base layer made of high bulk density
sandy clay (a soil particle density of 2.6 $gcm^{-3}$ with a porosity of 52%), and a middle
layer made of lower bulk density granules (a soil particle density of 2.6-2.8 $gcm^{-3}$ with a
porosity of 76%) (Kamai et al., 2008). The s-velocity structure observed in the base
layer was faster (harder) in comparison to the middle layer (Kamai et al., 2008).   For
the purpose of the archaeological studies, 6 trenches were excavated and landslide
remnants were observed in many of these trenches. The burial mound was originally
surrounded by a double moat, but most of this moat was buried in the past, and only a
part of it currently remains. The landslide deposits originated from the sediments in the
moat were dated, and the results were 1420-1510 AD with a method of the $C^{14}$ dating
(Sangawa and Miyazaki 2001). Since it is known that Fushimi Earthquake occurred in
1596, this burial mound collapse was thought to have been triggered by this earthquake
(Sangawa and Miyazaki 2001).



The top view of the scalps and landslides generated by the 1596 Fushimi Earthquake is
shown in Figure 1 (Kamai et al., 2008). The results of the trench survey indicated that
most of the landslide types were represented by a combination of translational and
rotational landslides (Kamai et al., 2008). Movement was inferred with the following
sequence: 1. the landslide mass moved near horizontally for a few meters, 2. the
transported landslide mass reached the inner moat, 3. the landslide mass slid down and
shifted from translational to rotational landslide mode. Conversely, It was found that an
exceptionally large-scale rotational landslide occurred in the northern region of the
round-shaped section of the burial mound (Scalps A and B in Figure 1). Whether the
burial mound deformation related to this rotational slide is connected to the translational
landslide had continued to be a mystery. The purpose of this work was to examine
whether muographically found evidence can be used to address this question.


Figure 1. Top view of Imashirozuka burial mound. Positions A and B indicate the



locations of the detectors for the current bidirectional muographic observations. The
shaded areas in red and blue indicate the viewing angle of each measurement. The solid
curves indicate the cross-sectional view of the mound at given elevation angles from
Positions A (red) and B (blue). The numbers indicate the elevation angles in units of
mrad.

In Figure 2, the cross-sectional view of the mound sliced along Line F in Figure 1 is
shown. This structure has been modeled based on the trench surveys conducted in 2008
(Kamai et al. 2008). Original surface of the mound (dashed lines in Figure 2) that was
estimated from the past archaeological studies, was lost by the landslide triggered by the
1596 Fushimi earthquake. The red lines indicate the slip surface of the landslide and at
the top of this surface, the existence of near-vertical cracks was expected. From these
trench surveys, the region indicated between the red lines and the solid black lines in
this figure was interpreted as the landslide mass, and displayed lower density than the
other part of the mound and thus, it was expected that muons could penetrate more in
this region (in particular at the top of this region).


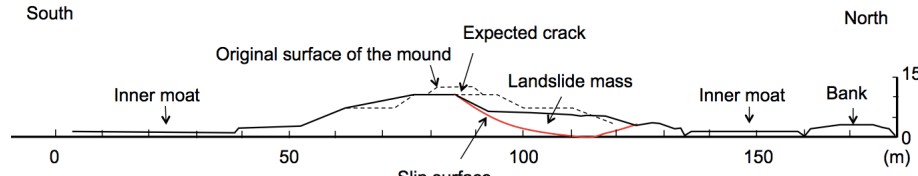

Figure 2. Cross-sectional view of the mound along Line F in Figure 1. The dashed lines
indicate the original surface of the mound, and the red lines indicate the slip surface of
the landslide triggered by the 1596 Fushimi earthquake. The authors drew this image
based on the work done by Kamai et al. (2008).

Mechanical fractures within rock and soil produce a significant amount of interparticle
space and these fractured zones are detected as lower-density-regions in muographic
images (Tanaka and Muraoka, 2013, Carbone et al. 2014). Likewise, when a landslide
occurs, various processes influence changes in the density distribution inside a burial
mound. When a crack is generated in the burial mound, the density is reduced along the
crack. If a large-scale collapse occurs, the collapsed landslide mass will contain a lot of
inter-particle voids, and the density will be reduced. If the geometrical arrangement is
altered between the high-density base layer and lower density middle layer due to the
ground motion such as a fault slip, the overall density distribution will be altered





accordingly. All of these variations can be imaged with muography.

**3. Method**
Bidirectional muographic measurements were conducted at the Imashirozuka burial
mound site so that the resultant images could be used for 3-dimensional interpretation of
the internal structure of the circular section of the mound. In particular, one of the
detector positions of the current bidirectional measurements were chosen on the
northern side of the round-shaped section (Position B) so that the area where the
extensive collapse occurred could be more closely observed. The positions chosen for
the current measurement are shown in Figure 1. The first measurement of the
Imashirozuka mound started at Position A on September 21, 2019. The data were taken
for 40 days, and subsequently the detector was moved to Position B to collect the data
for another period of approximately one month.

The detector employed for the current measurement was the multi-wire proportional
chambers (MWPC) based muographic observation system (MMOS) that consists of 6
layers of MWPCs and lead plates with a total thickness of 10 cm. A detailed description
of the MMOS can be found in elsewhere (Olah et al. 2018), and thus only the main
features are briefly introduced here. In between each of the MWPC, a 2-cm thick lead
plate accommodated in a 4 mm-thick-stainless steel case is inserted, thus the total
thickness of these radiation shields is equivalent to ~130   $gcm^{-2}$. These radiation
shields function as an absorber or a scatterer of low energy background particles that
include muons or other elecromagnetic particles. Only the straight trajectories
throughout 6 detectors are employed and recorded as muons. In the current
measurements, the total weight of the MMOS was 600 kg including the case, batteries
and gas bottle. The total power consumption of the detector was ~30W, and the six
400-Wh lithium-ion batteries loaded into the case allowed us the continuous operation
for 80 hours, and the recurrent charging and replacements of the batteries further
extended the time of the continuous operation. The flow rate of the Ar-$CO_2$ gas mixture
(Ar:80, $CO_2$:20) through the chambers was 1–2 liters per hour to enable continuous
operation for a few months with a standard 40L type (6,000 litters) gas bottle. The
casters attached to the bottom of the case facilitated movements of the detector around
the mound. Moisture absorbent boxes were equipped inside the box in order to retain
the humidity at a constant level around the MWPCs. The size of the active area of the
detector was 80 x 80 $cm^2$, and the distance between the uppermost and lowermost
stream detectors were 150 cm. The recorded muon tracks were stored and the numbers



of muon counts were directionally sorted out into a matrix with an angular binning
width of 8 x 8 mrad. As is indicated in Figure 1, the azimuthal viewing angle was
+/-500 mrad, however, due to the smaller geometrical acceptance for larger angles, only
the data within +/-400 mrad were used. The detector cost was ~60k US dollars, but the
operational cost was suppressed to a few thousand US dollars for entire operation
including the transportation, human resources for battery replacements and data
download.
Since the current target size is an order of 100 m, the following simplified analytical
expression can be applied for derivation of the relative density variations inside the
target volume because the muon's cutoff energy (the minimum energy of the muons that
can escape from the target volume) is much lower than the critical energy, 708 GeV in
$SiO_2$, the continuous ionization process is the main energy loss process (Tanaka and
Ohshiro, 2016).
$I_0/I_1 = (X_0/X_1)^{-\gamma}$,                                        (1)
where $I_0$ and $I_1$ is the remnant muon flux after passing through different densimetric
thickness of rock $X_0$ and $X_1$. The Greek symbol, $\gamma$, is the zenith-angular dependent index
of the power low of the integrated muon spectrum within 50-200 GeV. In this work,
only the "relative muon flux" was used for discussions of the density contrast inside the
mound. The obtained matrix has been normalized by the azimuthal distribution of the
open-sky flux so that the azimuthally angle-dependent acceptance has been canceled in
the image.
**4. Results**
Figure 3 shows the muographic image (Image A) taken at Position A that is indicated in
Figure 1. Corresponding azimuthal angles (-0.344 rad - 0.456 rad) are shown in Figure 1.
The distance between the detector and the peak of the mound was 70 m, and thus the
elevation angle of the mound peak was ~110 mrad (~6 degrees). Since the convex level
of the mound is small, the matrix was not re-binned in the elevational direction, but was
re-binned in 40 mrad in the azimuthal direction in order to increase the statistics. The
data were normalized to the azimuthal distribution of the open-sky muon tracks that was
unaffected by the existence of the mound, which corresponds to the elevational region
between 300 and 360 mrad in order to derive the "relative muon flux". The bottom right
green-colored region in Figure 3, where the number of muons was counted less than
other regions corresponds to the direction, where the background mound (rectangular
section) was overlapped to the foreground mound (circular section); hence longer path
lengths for muons, as can be seen in Figure 1. The reddish patch that can be seen on the
left side of this green-colored region indicates a low-density collapsed landslide mass
on the northern slope of the mound. It was expected that the region around scalps
(arc-shaped lines in Figure 1) had cracks, and thus the average density along these
cracks was significantly lower than the density around these. This density reduction
effect is maximized in muographs when the muon's ray path is parallel to these cracks.
From Position A, this direction corresponds to the azimuthal angular range between 200
mrad and 300 mrad (see the position indicated by "Crack A" in Figure 1).

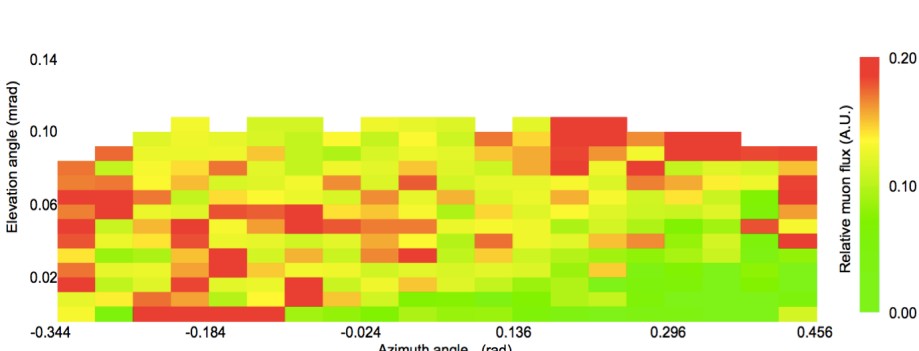

Figure 3. Angular distribution of the relative muon flux, as was observed from the
measurement at Position A. The horizontal and vertical bin widths are respectively 40
mrad and 8 mrad. The azimuthal distribution of the relative muon flux was normalized
to the total number of muons counted at each elevation angle.
Figure 4 shows the azimuthal distribution of the relative muon flux at shallow depths (at
elevation angles of 108 mrad (Figure 4A) and 100 mrad (Figure 4B)). The solid lines
are the expected muon flux. These lines were drawn based on the geometrical thickness
of the mound along the muon paths (Figure 1) by assuming the uniform density
distribution inside the mound. In these three images, the following three features can be
found. (A) Overall, the excessive flux of muons was observed in the positive azimuthal
angle region. This indicates that the average density in the positive azimuthal angle



region is lower than that in the negative angle region. An overall density variation
between them is 10-20%. (B) A strongly excessive muon flux can be found in the
azimuthal angle region between 176 mrad and 296 mrad in Figures 4A and 4B. The
statistical significance was more than 4σ. (C) In Figures 4A, there is also a low-density
region within the azimuthal angle range between 296-456 mrad. The position of this
low-density region corresponds to that of Trench F (dotted lines in Figure 1). From (A)
and (B), it was inferred that a large almost vertical crack exists in the shallow region,
however its existence was not clear when it's deeper than 2 m because of the effect of
overlapping the rectangular-shaped background mound (see the green-colored area at
the right bottom region of Figure 3). The density variations of this possible crack were
20-30% in comparison to the average density of the other part of the mound. The crack
width was 80-120 mrad that was equivalent to 6-8 m when considering the distance
between the detector and Crack A of 70 m.



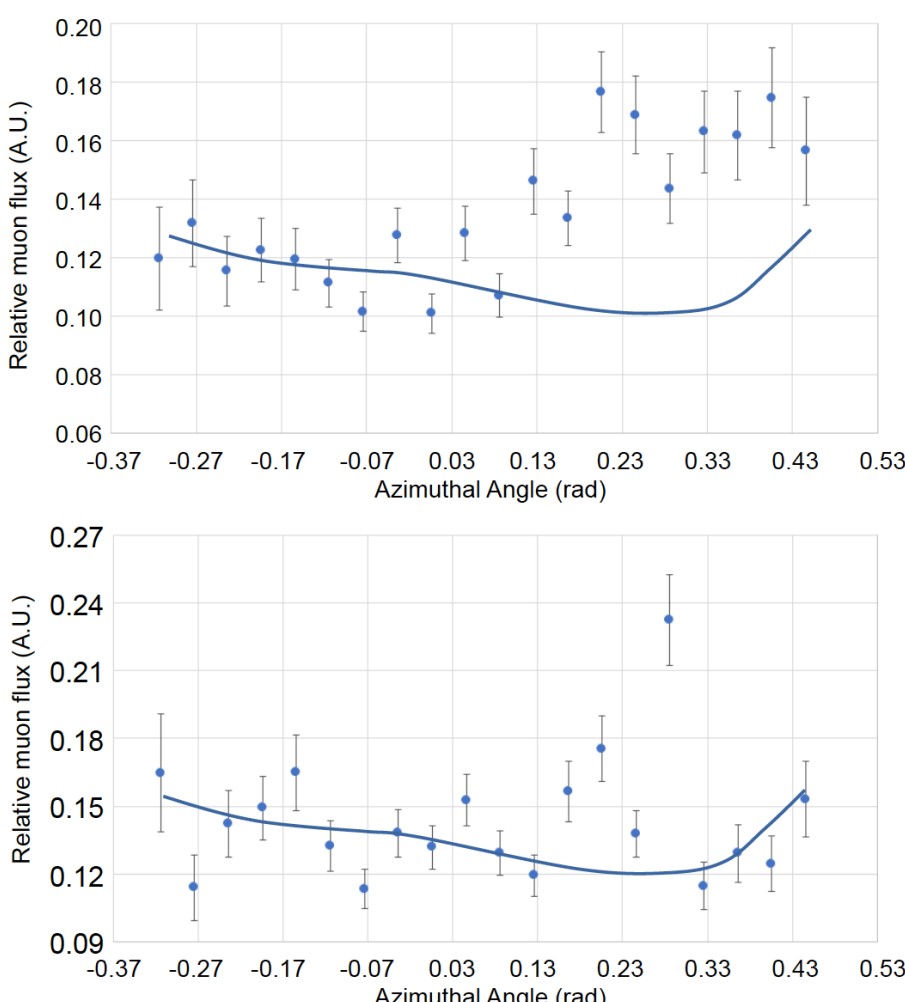


Figure 4. Azimuthal distribution of the relative muon flux for elevation angles of 108 mrad (A), 100 mrad (B) and 92 mrad (C). The solid curves indicate the expected horizontal muon flux variations.

306 Crack A was not parallel to the muon's ray path at Position B (Figure 1), however, 307 Crack B was parallel to those in the azimuthal angle range between 300-420 mrad. 308 Therefore, it was expected that the similar structure to Crack A would be observed in 309 this angular region.    Figure 5 shows the muographic image (Image B) taken at

Position B. Since the distance to the mound peak (50 m) was closer at Position B, almost doubly defined density structure was imaged. The data were normalized to the azimuthal distribution of the muon tracks recorded within the elevational range between 300 and 360 mrad in order to derive the "relative muon flux". Corresponding azimuthal angles (-0.376 rad - 0.424 rad) are shown in Figure 1.

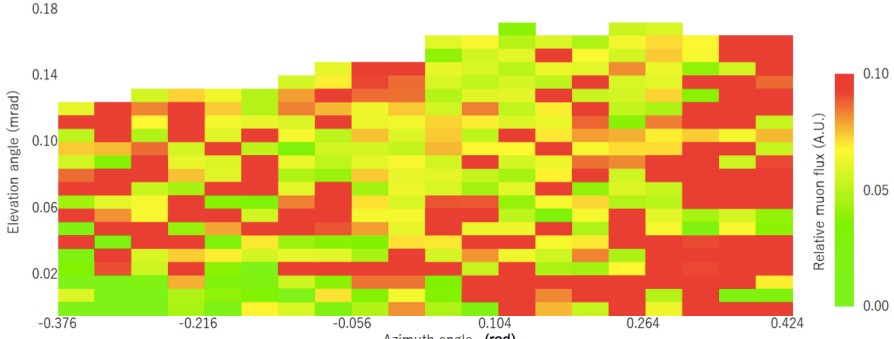

Figure 5. Angular distribution of the relative muon flux observed at Position B. The horizontal and vertical bin widths are respectively 40 mrad and 8 mrad. The azimuthal distribution of the relative muon flux was normalized to the total number of muons counted at each elevation angle.

In Figure 6, the azimuthal distribution of the relative muon flux for elevation angles of 68 mrad - 172 mrad are shown. In these images, the excessive muon flux was found within the azimuthal angle range between 264-424 mrad. The statistical significance was overall more than 1σ. This low-density region was interpreted as the crack associated with the same scalp (Scalp A), and it was found that the vertical extent of the crack was much deeper than what could be seen in Image A. The crack width was at least 80-160 mrad that was equivalent to 4-8 m when considering the distance between the detector and Crack B. The reddish region in Figure 5 that can be seen on the left side of Crack B indicates low-density collapsed landslide mass, with a mixture of the remnant of the past excavation at Trench F.

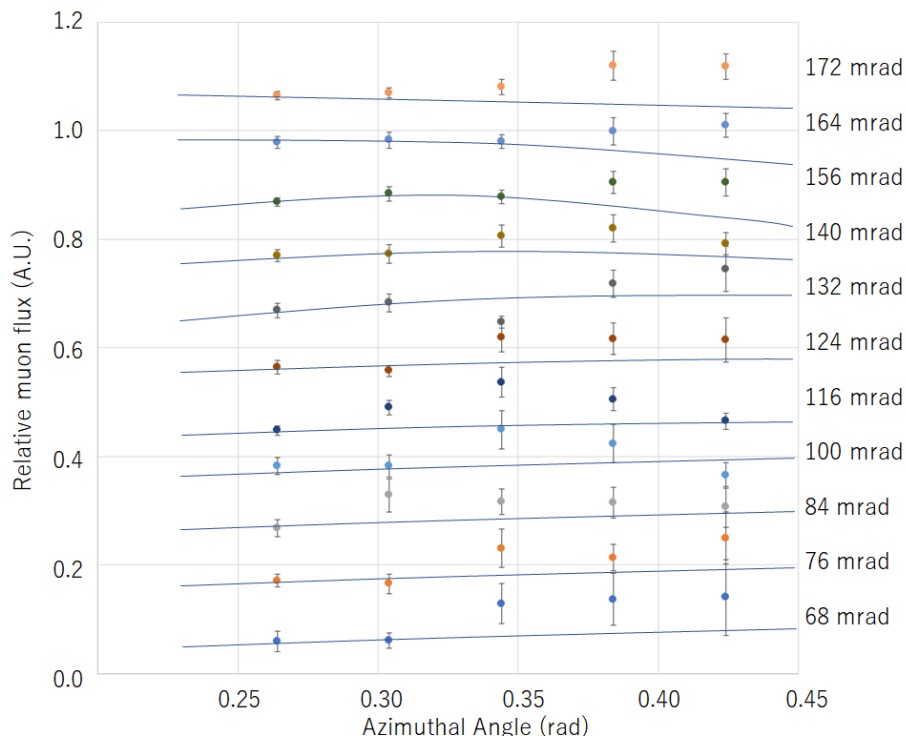

Figure 6 Azimuthal distribution of the relative muon flux for various elevation angles.
The solid curves indicate the expected horizontal muon flux variations.

**5. Discussion**

From the bidirectional muographic images taken in the current measurements, the
following interpretations were derived.

(A) The reddish patches that can be seen on the left side of Images A and B indicate an
existence of a large-scaled collapsed landslide mass on the northern slope of the mound.
The collapsed landslide mass has a significant amount of the interparticle space; hence a
lower density in comparison to the surrounding regions.

(B) The vertical low-density regions at the top of the mound in Images A and B show
that there is a large-scale vertical crack behind Scalp A. The widths of these vertical
cracks were both 4-8 m thus it is reasonable to assume they are associated with the same

scalp.

In conclusion, the following picture was proposed. In prior trench-survey-based works,
most of the landslides that deformed this burial mound structure were found to have
been caused by a translational process. On the other hand, there was an exceptionally
large-scale rotational slide was found in the north region of the round-shaped section of
the mound, and the stone chamber was deformed and destroyed by this collapse process.
However, in the current muographic observations, a large-scale vertical crack was
discovered at the top of the round-shaped section, and it was found that the burial
mound deformation that connected to the translational collapse process also occurred
behind this rotational landslide. These data indicate that there was an intrinsic problem
with the stability of the basic foundation of the Imashirozuka mound before the 1596
Fushimi Earthquake.  Changes in the foundation as a response to shaking from the
earthquake may have produced this large-scale burial mound collapse.

The burial mound seems to have a robust structure, more stable against earthquakes
than slender buildings like clock towers. However, a number of the ancient burial
mounds throughout Japan have collapsed from earthquakes and many modern buildings
are now built upon them. A small fraction has survived since early times, however, they
do not always indicate the earthquake-free sites. They represent an example of the final
designs of ancient Japanese construction since they have remained even after having
experienced a number of destructive earthquakes.

The technique of muography, which can probe seismically damaged ancient mounds is
similar to medical radiography which seeks to find the position, formation, and size of
the fractured zone inside the human body. In general, it is difficult to understand the
extent of damage, for example, of a patient's external wound without also understand
what is happening inside the body. The outside structure of ancient mounds is similar.
The surface of them has usually been naturally or artificially eroded with added
vegetation covering the shape during a long period of time it has existed. However, the
inside is more intact. For this reason, the trench survey technique (physically digging a
trench into the structure) to understand the "inside" can reveal valuable data.
However, similar to the manner in which x-ray photographs are usually applied to a
diagnosis before surgery is considered, muography is a more convenient and
noninvasive technique to effectively understand the overall inside structure to assess the
effect of time and natural disasters on the structure as a whole.




The current proof of concept measurement has attempted to show whether the technique
of muography increases the possibilities of finding more physical evidence related to
past earthquakes by selecting the Imashirozuka mound as an example. Obviously, the
specific earthquake damage of each burial mound is unique and cannot be generalized.
Its response depends not only on its material properties of the mound including its
mechanical properties of its foundations (strength and rigidity), but also on the ground
motion during an earthquake. Surveying and mapping various mounds that are thought
to be affected by the earthquake will provide a valuable data for us to verify and sort out
the factors that caused the damage.

Not only Imashirozuka mound but also other various burial mounds including the
Mishima mound group and the Kobo mound group are concentrated along the Rokkou
active fault system and its next neighbor Arima-Takatsuki tectonic line. The current
muographic results suggest that a combination of muography and the techniques of
trench survey or other conventional geophysical techniques can contribute towards the
construction of a more comprehensive understanding of the seismic response and
deformation of each burial mound. The characteristics of muography would allow
researchers to conduct an investigation of several sites quickly and efficiently to grasp
the general trend of the ensemble of these sites.   Incorporating the muography
visualization technique into engineering expertise and in conjunction with historical
comparanda would utilize a new potential:   by acquiring this new, valuable data from
these ancient burial mounds in Japan and other similar sites worldwide, we would
increase our ability to tackle future challenges of natural disaster preparation.

**Acknowledgements**
The authors acknowledge Toshitaka Kamai for valuable discussions about the current
muographic observation results. The authors also acknowledge Takefumi Hayashi for
his coordination and support the current measurements, Fumitaka Yoneda & Chikara
Inoue for their valuable archaeological advice, Ichiro Kanegae for provision of past
excavation research materials of Imashirozuka mound, and Masao Uchida for his
support as a chief administrator of Imashirozuka park.

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
