# Peer review of "Muography as a new tool to study the historic earthquakes"

_Geoscientific Instrumentation, Methods and Data Systems, 2020_

## Referee Comment (RC1) · Anonymous Referee #1 · 23 Jun 2020

General comments. The article describes the study of an ancient burial mound with the muography technique in order to give an interpretation about ancient techniques used for the construction of these kinds of buildings. In this sense the article aims to propose muography as a new noninvasive tool for the archeology. Results, performed from two different observation points, show the presence of low density regions interpreted as collapsed landslide mass and vertical cracks. The reported conclusions, that takes in account also previous trench-survey-based works, about an intrinsic problem with the stability of the basic foundation of the mound are not, in the opinion of the reviewer who is not an expert of the sector, immediately clear.

[Figure]

The article is well written, and the scientific interest is notable since the technique is for the first time proposed for this kind of application. Some minor comments and technical corrections are proposed.

Specific comments. 142: the word scalps is not clear to me. I consulted some English dictionaries and the meaning I have found is not appropriated for the context. Please specify better what are you describing.

210-212: an indication of the approximative value of the energy cut on muons and e.m. particles would be appreciated.

223.224: could you provide the detector spatial and angular resolutions ?

226: the 8x8 mrad(^2) angular binning corresponds to the angular resolution ? see also previous comment. A mistyping is also reported in the technical corrections section.

254: Since the convex level of the mound is small [. . .] Could you clarify the meaning ?

260: An indication of the total number of muons collected and of the contents of muons recorded in the bins could be appreciated.

261: could you better describe what are the background and foreground mound and the effect on the measurement ?

263: I think that it is not appropriate to claim here a lower density region since no normalization to the effective thickness of material crossed by muons have been applied to the plot of Figure 3.

311: almost doubly defined density structure was imaged. It is not clear to me the sense. Please describe better.

347: do you have an estimation of the percentage variation of the density ?

Technical corrections

226: 8x8 mrad -> 8 mrad x 8 mrad

Figure 3: vertical axis unit is in rad and not in mrad.

[Figure]

---

## Referee Comment (RC2) · Anonymous Referee #2 · 3 Jul 2020

General comments

The paper describes the application of muography, a field which developed considerably in the recent decade, on a very interesting target: identifying remnant structures from an earlier earthquake on a burial mound. The subject widens the scope of muography, and helps the community with better understanding both the methodology, as well as the scientific possibilities offered for archeological studies. The results are clearly worth publishing.

Specific comments

Fig. 1 is a very relevant figure for the measurement environment, however it is too

dense. E.g. l.150 says Scalps A and B in Figure 1, which is not identifiable to me. Figure caption refers to "solid curves", which are probably with matching color with the viewing directions (red and blue), but otherwise there are a lot of solid curves on the figure (e.g. landslides). It would be important to make the captions precisely matching with the figure (e.g. the image is now gray below the explanatory lines and writing; on the left bottom caption "Landslides" are with gray shaded area and black line. Figure has no black lines. (also later l.266 says "scalps (arc-shaped lines in Figure 1)".)

It may be a possibility to split the image into two, one more for the existing geometry, the other for the interpretation (indicacing Cracks / Scalps, observation direction elevations, etc).

The interpretation of the geometry is a bit complicated, in the text with references to Trench F, Crack A and B, Scalp A, etc. These may be indicated on the relevant muogram images?

l.326 argues for significance "was overall more than 1 sigma", which is not too convincing. To me it looks more like 2-3 sigmas, in multiple independent measurement points.

l.325 says "angle range between 264-424 mrad", which seems the combination of Crack A and B, however, l.327 says "associated with the same scalp (Scalp A). Please clarify.

Technical comments

There seems a confusion on figure numbering, now there are two different Figure 3-s, probably the colored muogram around l.273 is Figure 3, and around l.300 (Azimuthal distribution...) is Figure 4. On this latter, indicate panels A, B and C.

l. 323 refers to Figure 6, which is non-existent, but must be the Figure 5.

Please check carefully the figure numbering and its consistence in the text references.

l.20: "... recorded in this historical heritage sites." Here "sites" should be singular (heritage site).

(Introduction: the word "however" is used a bit too often (l.31, l.33, l.40, l.62), breaking the argumentation line. Consider replacing some by re-wording, if seems appropriate)

l.148: Conversely, It was... ("it" should not be with capital)

---

## Author Comment (AC1) · 10 Jul 2020

The authors acknowledge two anonymous referees for their valuable suggestions. The point-by-point revision notes are appended below.

Replies to Anonymous Referee #1

(1) Specific comments. 142: the word scalps is not clear to me. I consulted some English dictionaries and the meaning I have found is not appropriated for the context. Please specify better what are you describing.

The word "Scalps" was replaced with the landslide headscarp

(2) 210-212: an indication of the approximative value of the energy cut on muons and e.m. particles would be appreciated.

We added the following sentences to the manuscript. The penetration of muons and electrons were simulated in GEANT4 simulation framework (Olah, L. et al., 2019). The analysis cut on the goodness of track fit was set to 1.5 to suppress the penetration of muons down to 10 % those had the energy of < 1 GeV. This simulation study showed that the electromagnetic component did not create signal in the MMOS.

(3) 223.224: could you provide the detector spatial and angular resolutions ?

226: the 8x8 mrad(ЁĘ2) angular binning corresponds to the angular resolution ? see also previous comment. A mistyping is also reported in the technical corrections section.

We added the following sentences to the manuscript. The wire distances were designed to be 12 mm in MWPC detectors to provide a fair positional resolution of approx. 4 mm even if lead plates were applied between the MWPCs (Varga, D. et al., 2015; Varga, D. et al., 2016; Olah, L. et al., 2018). The angular resolution of 1.5 meter-length tracking system was approx. 2.7 mrad (Olah, L. et al., 2018).

(4) 254: Since the convex level of the mound is small [. . .] Could you clarify the meaning ?

We rephrased the text as follows. Since the aspect ratio of the mound, i.e., the ratio of its width to its height (10:1) was large,

(5) 260: An indication of the total number of muons collected and of the contents of muons recorded in the bins could be appreciated.

We added the following sentences to the manuscript. The total number of muons collected at Position A in the elevational-angle-region below 180 mrad was 76,682. The number of muons recorded in the bins at an azimuthal angle of 0 ranged from 30 to 500, depending on the elevational angle.

The total number of muons collected at Position B in the elevational-angle-region below 180 mrad was 15,214. The number of muons recorded in the bins at an azimuthal angle of 0 ranged from 15 to 100, depending on the elevational angle.

(6) 261: could you better describe what are the background and foreground mound and the effect on the measurement ?

We rephrased the text as follows. The bottom right green-colored region in Figure 3, where the number of muons was counted less than other regions corresponds to the direction because in the positive azimuthal angular region at Position A, the rectangular section of the mound provided the additional path length for muons that arrived at lower elevation angles.

(7) 263: I think that it is not appropriate to claim here a lower density region since no normalization to the effective thickness of material crossed by muons have been applied to the plot of Figure 3.

We removed the following sentences from Results section.

The reddish patch that can be seen on the left side of this green-colored region indicates a low-density collapsed landslide mass on the northern slope of the mound.

Also, we removed the following related sentences from Discussion section.

(A) The reddish patches that can be seen on the left side of Images A and B indicate an existence of a large-scaled collapsed landslide mass on the northern slope of the mound. The collapsed landslide mass has a significant amount of the interparticle space; hence a lower density in comparison to the surrounding regions.

The claim in Discussion section will not be affected by removing these sentences.

(8) 311: almost doubly defined density structure was imaged. It is not clear to me the sense. Please describe better.

We rephrased the text as follows. Since the distance to the mound peak (50 m) was

closer at Position B, the spatial resolution at the mound peak was improved for a given angular resolution of the tracker.

(9) 347: do you have an estimation of the percentage variation of the density ?

Related to the comment (7), since it was not appropriate to claim a lower density region since no normalization to the effective thickness of material crossed by muons have been applied to the plot of Figure 3, this part was removed from the manuscript.

(10) 226: 8x8 mrad -> 8 mrad x 8 mrad

Corrected.

(11) Figure 3: vertical axis unit is in rad and not in mrad.

Corrected.

Replies to Anonymous Referee #2

(1) Fig. 1 is a very relevant figure for the measurement environment, however it is too C1 GID Interactive comment Printer-friendly version Discussion paper dense. E.g. l.150 says Scalps A and B in Figure 1, which is not identifiable to me.

"Scalps A and B" were removed from the sentence. Instead, we rephrased it to "large-scale rotational landslide occurred in the north side of the round-shaped section of the burial mound." so that the reader can recognize them from Figure 1.

(2) Figure caption refers to "solid curves", which are probably with matching color with the viewing directions (red and blue), but otherwise there are a lot of solid curves on the figure (e.g. landslides).

It may be a possibility to split the image into two, one more for the existing geometry, the other for the interpretation (indicacing Cracks / Scalps, observation direction elevations, etc). The int

We added an inset to Figure 1 so that the geometrical information is separately given
in the figure. The caption was modified so that the indication of the solid curves could be more distinctive.

(3) It would be important to make the captions precisely matching with the figure (e.g. the image is now gray below the explanatory lines and writing; on the left bottom caption "Landslides" are with gray shaded area and black line. Figure has no black lines. (also later l.266 says "scalps (arc-shaped lines in Figure 1)".)

We modified the color of the arc-shaped lines and trench marks in the legend so that they matched with the ones in the topographic drawings.

(4) l.326 argues for significance "was overall more than 1 sigma", which is not too convincing. To me it looks more like 2-3 sigmas, in multiple independent measurement points.

We added a phrase. "The statistical significance was overall more than $1\sigma$ïĂňïĂăïĄćïĄţïĄŕïĂăwas increased to 2-3$\sigma$ in the shallower region of the mound."

(5) L.325 says "angle range between 264-424 mrad", which seems the combination of Crack A and B, however, l.327 says "associated with the same scalp (Scalp A). Please clarify.

We rephrased the sentence. "This low-density region was interpreted as the combination of Cracks A and B"

(6) There seems a confusion on figure numbering, now there are two different Figure 3-s, probably the colored muogram around l.273 is Figure 3, and around l.300 (Azimuthal distribution...) is Figure 4. On this latter, indicate panels A, B and C. l. 323 refers to Figure 6, which is non-existent, but must be the Figure 5. Please check carefully the figure numbering and its consistence in the text references.

Corrected.

(7) l.20: "... recorded in this historical heritage sites." Here "sites" should be singular

(heritage site).

Corrected.

(8) (Introduction: the word "however" is used a bit too often (l.31, l.33, l.40, l.62), breaking the argumentation line. Consider replacing some by re-wording, if seems appropriate)

The second however was replaced with other wording so that this word appears less frequently.

(9) l.148: Conversely, It was... ("it" should not be with capital)

Corrected.